# Unusual Micro Carbon Rods Formed from PET Plastic via Pyrolysis and Annealing in CO₂/He Co-Gas

Yi'en Zhou and Liang Hong *

Department of Chemical & Biomolecular Engineering, National University of Singapore, 4 Engineering Drive 4, Singapore 117585, Singapore; yien.zhou@ontoinnovation.com
* Correspondence: chehongl@nus.edu.sg

**Abstract:** This study investigates the transformation of activated carbon (AC) powder, derived from polyethylene terephthalate (PET) through pyrolysis, into a specific type of short cylindrical carbon. This carbon-to-carbon (C-C) transformation was completed by annealing the AC powder in a co-gas atmosphere of He and $CO_2$. This produces low-porous, amorphous, and micro carbon rods (MCR) in micron size. It is suggested that a so-far unknown growth mechanism originates from the oxidation role of $CO_2$, initiating the curving of polycyclic aromatic hydrocarbons (PAHs) sheets. This annealing step was followed by layer-by-layer sheet stacking steps to render the thick rods. This thickness is also created by the simultaneous occurrence of rare carbon nanotubes, supposedly formed initially from curling a small sheet of PAH surrounding carbon nanoparticles to create a tube template for subsequent cylindrical growth. This is the first example of CNT growth through C-C transformation rather than the other vapor deposition routes. As the main product, MCR is amorphous and fairly porous, with an average aspect ratio greater than 10, which possesses potential applications as a mechanical reinforcing or energy-attenuation filler for different composites.

**Keywords:** PET thermoplastic; carbon rods; layer stacking; curvature of PAHs; pyrolysis; carbon-to-carbon transformation

## 1. Introduction

The industrial production of long carbon fibers [1] has always been achieved through the carbonization of strands of polyacrylonitrile, of which the conversion mechanism from polymer to nitrogen-containing graphite structure, through expansion and alignment of graphite planes along the axial direction, is well comprehended. Following this fibrous polymer-to-carbon strategy, recent progress has demonstrated electrospinning technology's success in producing polymer-derived carbon nano or micron fibers with an inbuilt porous structure [2–11]. In contrast to carbon fibers, carbon nanorods with much smaller diameters, made by a chemical vapor deposition technique, have also emerged [12–14]. From the application perspective, low-cost starting materials and simple production technologies to prepare carbon fibers are appealing for various composites. Currently, refinery by-products and plastic wastes have great potential which should be explored. Relative to plastic wastes, petroleum distillation residues contain significant portions of heavy aromatics and are close to achieving the graphitic structure. Fundamentally, propagating sp² carbons along the axial direction under an annealing condition is the focus, which usually requires a precursor in a fibrous shape as well as a high carbon-to-hydrogen ratio. However, investigating the fabrication of carbon nano or micron fibers utilizing irregular carbonaceous substances to undertake carbon-to-carbon (C-C) transformation has rarely been initiated. A publication in the late 1990s reported such an effort [15]. This attempt is attractive because it could allow plastic wastes to be converted into MCR through a carbonaceous intermediate for composite filler. PET is one of the most common plastic wastes; converting it into micro carbon rods could be an effective way to use it [16]. Although it is difficult to generate

carbon rods with a very high aspect ratio using C-C conversion, MCR must possess superior mechanical strength and a more regular internal structure than its carbonaceous precursor.

Accordingly, the cyclic polyaromatic hydrocarbons (PAHs) species are considered the essential carbon blocks to participate in a C-C transformation at high temperatures. This study investigates a specific change in the activated carbon powder, i.e., a carbonaceous substance derived from PET pyrolysis, to low porous, amorphous, short carbon micron rods. The electron microscopic observation suggests that the growth of MCR relies on using curved PAHs to undergo the layered assembling transversely. The curving of PAHs structurally relies on degenerating the p-conjugation extent by including distorted carbon hybridization geometries. This attribution is inspired by sparse multi-walled carbon nanotubes (MWCNT) in each C-C product batch. These CNTs are supposedly formed initially from the curling of small PAH sheets of PAHs surrounding a carbon nanoparticle under rare conditions. It is the first example of CNT developed by non-catalytic and non-CVD approaches. PET is a unique precursor leading to MCR through the above steps, presumably because of its alternating ester and phenylene linkages.

Furthermore, to understand whether PET has potential for the eventual generation of MCR, we used polypyrrole (PPy) as a control because it has a conjugate backbone and is more ready to assume extensive p-conjugated systems relative to PET. Nevertheless, PPy cannot afford the sheet-to-rod transformation under the same C-C conversion conditions. This observation suggests the impact of polymer precursor on the chemical reactivity of its inherited PAHs. The evolution of MCR could only be viable with those PAHs that permit the degeneration of the $sp^2$ symmetry of carbon under designated treatment conditions. The PPy-based PAHs do not demonstrate this kind of chemical reactivity. For the PET-derived AC, the C-C transformation involves the oxidation of a number of $sp^2$ carbons by $CO_2$ to form ketene and ether at the annealing temperature so that the organic functional groups prompt the curvature of PAHs. This study initially aimed to prepare highly porous AC on the micron scale as an adsorbent in a randomly packed bed to remove soluble and insoluble organics from wastewater streams [17]. We chose PET as the starting material because it is a significant component of plastic waste. It was then found that MCR and CNT formed through optimizing the pyrolysis of PET and then annealing the resulting AC in the co-gas atmosphere. The MCR obtained is inadequate to be an adsorbent due to its lower surface area; it could be used as a structural filler. On the other hand, this study reveals an inverted BET isotherm, implying the adaptability of the assembled carbon layers. This trait could be related to the attenuation application to damp mechanical waves of certain frequencies because of the adjustable gaps among the graphitic layers in MCR [18]. Therefore, MCR is predicted to function as an energy-damping medium.

## 2. Materials and Methods

### 2.1. Preparation of PET-Derived Activated Carbon (PET-AC) Powder

The PET polymer (Sigma Aldrich, St. Louis, MO, USA) was the starting material in the preparation of AC. First, the PET polymer (10 g) was placed in the middle portion of a quartz tube (diameter: 50 mm; length: 1200 mm), and the reactor was purged with an Ar stream (585 $cm^3 \cdot min^{-1}$) for 10 min. The polymer precursor was then subjected to pyrolysis at 400 °C using a heating ramp rate of 5 °C $min^{-1}$ under the Ar atmosphere. The pyrolysis then proceeded for an hour. After that, the carbonaceous substance formed was soaked for 1 h in $CO_2$ (700 $cm^3 \cdot min^{-1}$) at 1 atm and 700 °C using a heating ramp rate of 5 °C $min^{-1}$ from 400 °C. After this activation step, the reactor and product were allowed to cool to room temperature under an Ar purging stream. The AC was then crushed into a fine powder using a mortar and pestle before washing the powder multiple times in deionized water to remove the soluble impurities and centrifuged. Finally, the AC powder was filtered and dried overnight in an oven.

### 2.2. Formation of Micro Carbon Rods (MCR) via the C-C Transformation

The AC powder was packed into a micro-tubular reactor, fed with a co-gas stream of He-CO$_2$ (*vol./vol.* = 1 flow rate), and heated to 800 °C (1 atm) for 5 h, i.e., an annealing step. After the treatment, the reactor was cooled to room temperature under a helium-purging flow. The carbon samples of this process were characterized on a transmission electron microscope (TEM, JEOL 2100 LaB6) and a field electron scanning electron microscope (FESEM, JSM-6700F, JEOL). Subsequently, the phase compositions of the carbon samples were examined using an X-ray diffractometer (Brucker D8 Advance, Cu K$\alpha$ radiation $\lambda$ = 1.54 Å). The porous features (BET surface area and pore size distribution) of the carbon samples were analyzed by the BET and BJH methods [19] on Autosorb (Autosorb-1, Quantachrome instruments). Finally, the two carbon samples were scrutinized using infra-red spectroscopy (FT-IR, Bio-Rad Excalibur FTS-3500 FT-IR spectrometer) using the KBr pellet window method to determine the organic functionality at the above two conversion stages.

## 3. Results

### 3.1. The Consecutive Conversion Steps from PET Plastic to MCR

According to recent studies on PET pyrolysis [16,20], the pyrolysis of PET generates several types of oxygen-containing fragments and compounds in the volatile product stream. This implies that the main chain of PET underwent extensive cleavage within a short time interval, resulting in numerous small PAHs, primarily owing to the condensation of the phthalate units. The resulting PAHs structurally possess more obvious edge characteristics of aromatic rings, typically the C-H bending. In our study, the IR spectrum of AC (Figure 1) shows an apparent absorption band at 1887 cm$^{-1}$, attributed to the overtones of the strong aromatic C-H bond out-of-pane bending [21]. Except for this characteristic, both absorption bands at 2072 cm$^{-1}$ and Ar-O (1214 to 1010 cm$^{-1}$) are attributed to ketene and ether groups, respectively. Moreover, PET-AC exhibits a stronger aromatic ether characteristic than MCR, proposing that the ether linkages of the former contribute to the growth of graphitic sheets in the subsequent annealing step. Compared to MCR, PET-AC also presents a more apparent IR bump at 1597 cm$^{-1}$ due to the carbon skeleton stretching.

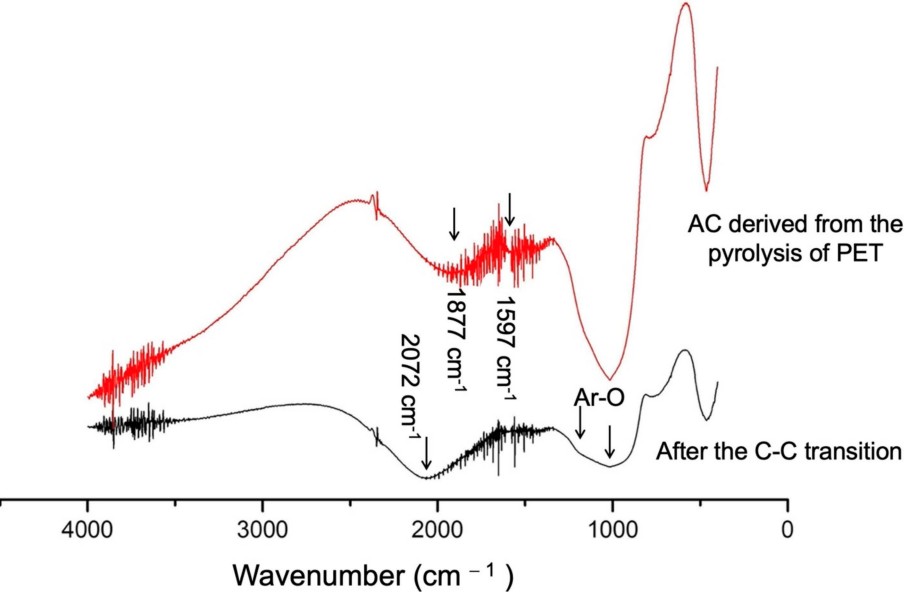

**Figure 1.** FT-IR spectra of the PET-AC and the carbon specimen obtained from the C-C transformation under co-gas soaking.

According to the electron microscopic image of PET-AC (Figure 2), an arbitrary AC particle presents a fine structure consisting of nanograins with sizes of mainly 30 to 40 nm (inset). It is important to note that the pyrolysis of PET was first carried out at 400 °C;

this pyrolysis temperature sustained only a mild decomposition rate according to the TGA study [22], which shows that PET undertakes swift mass loss at 450 °C. Conducting the pyrolysis at 400 °C averted fast and chaotic crosslink of thermally more vulnerable carbonaceous substances still owning higher H/C ratios. Hence, this allowed the evolution of relatively smaller PAHs, and even PAHs in the first place. The subsequent calcination at 700 °C in $CO_2$ drove, on the one hand, the 2D expansion of these PAHs via aromatic condensation [23], but also implemented reactive sites (e.g., secondary or tertiary C) in the PAHs due to the etching effect of the reverse Boudouard reaction.

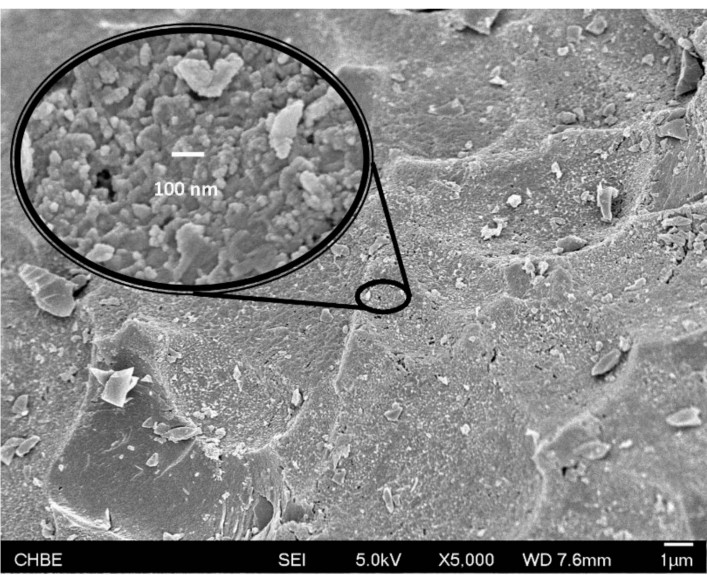

**Figure 2.** FE-SEM image of a PET-AC particle surface and the inset displays the fine structure of the particle.

These nanograins are, in principle, aggregates of different PAHs with diameters of several dozen nanometers generated during the pyrolysis. When carefully examining the fine structure of the individual nanograin of PET-AC, we observed Cn carbon cages attached to short carbon nanofibers that have solid bulk (Figure 3). Namely, these carbon cages are structurally constructed by nano carbon balls (<10 nm); rationally, they are fullerenes. They were formed from the edge association of those twisted, bent, or curved PAHs with smaller sizes, owing to containing some $sp^3$ sites in the carbon skeleton of the conjugated system. In the meantime, the rare solid nanofibers evolved from the deposition of the distorted PAHs together with other small enough carbonaceous species. The growth of fullerene-like species, however, ceases after the co-gas annealing. Ostensibly, the co-gas, on the one hand, acts as a cage-opening reagent and simultaneously encourages curved PAH sheets' development.

When the PET-AC particles were subjected to annealing in a stream of helium and carbon dioxide, i.e., the co-gas, at 800 °C, it gave rise, ultimately, to abundant short MCR (Figure 4). The inset of this figure exhibits a defect spot on the surface of an arbitrarily selected MCR for scrutiny, in which a top layer was peeled off, revealing a superficially dense core. In addition, on the inner side of the peeled layer, there are traces of nano bits (marked by a dashed circle) denoting the residues of AC nanograins left behind from the C-C transformation. Similarly, such un-completely reacted AC nanograins can be spotted on the rest locations of the selected rod. It is also noteworthy that besides MCR, the debris has curved brinks, indicating the geometries of the intermediates occurring through the C-C conversion. These intermediates reflect the assemblies of curved PAH sheets. Lastly, the MCR has an aspect ratio (L/D) of at least 10. The rod exemplified in Figure 4 has an aspect ratio greater than 16.

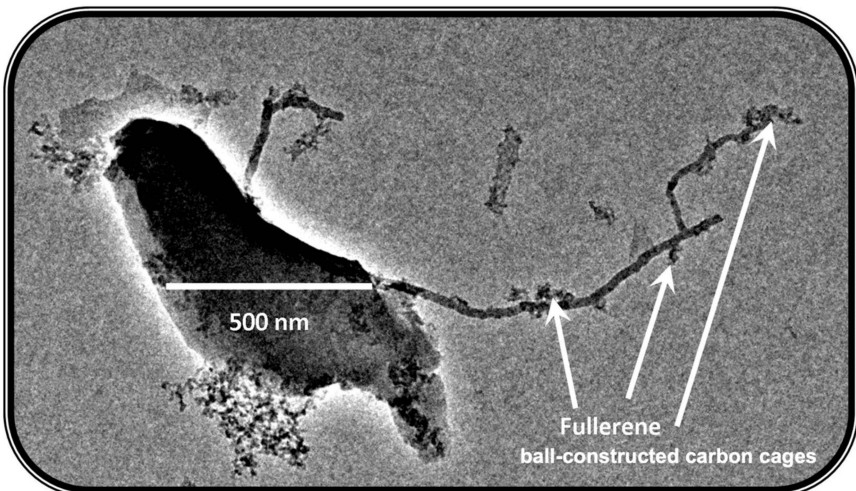

**Figure 3.** The PET-AC contains agglomerated carbon cages along the fibers formed.

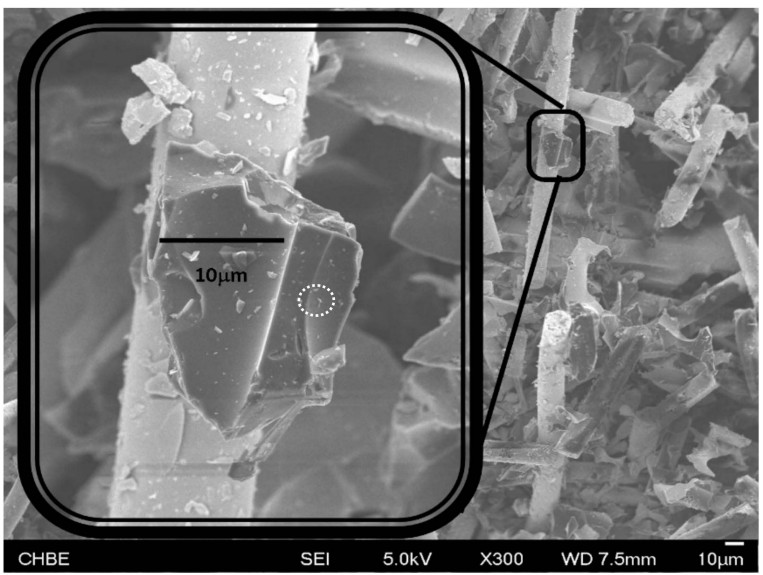

**Figure 4.** The FE-SEM image of the C-C transformation product.

Considering that PAHs play a vital role in the growth of MCR, it is therefore intriguing to explore whether a polymer precursor having a more extensive conjugated carbon frame could result in a better version, in terms of length, surface cleanness, etc., of MCR through the same dual heat-treatment steps. Polypyrrole (PPy) was chosen for this examination because it readily generates PAHs upon pyrolysis [24,25]. Their TGA analyses show that PPy is thermally more fragile than PET [22], likely due to its polar conjugated structure. PPy, once undergoing thermal cracking, should have a stronger tendency to achieve high conjugation scopes through the ready assembling of its $sp^2$ fragments. As a result, the PPy-derived AC carbon flakes are densely packed due to intense p-stacking propensity. The resulting carbonaceous bits, therefore, lack the required reactivity (with $CO_2$ to de-generate the p-conjugation extent) (Figure 5) to lead to curved PAHs sheets under the co-gas treatment. As a result, only a few carbon fibers developed. The rest remain as dense chunks, implying an overwhelming random planar aggregation of PAHs instead of curling up under the C-C annealing conditions.

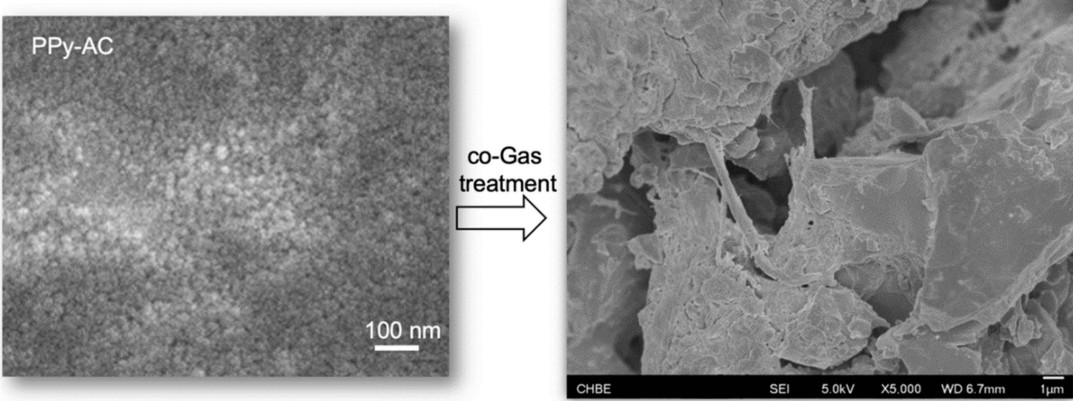

**Figure 5.** The C-C transformation of PPy-derived AC, a microscopic examination.

### 3.2. The Surface and Bulk Features of MCR

Surface analysis was performed on the above two PET-based carbon specimens aiming at their surface areas and pore volumes (Figure 6). The $N_2$-adsorption capabilities reduce with the increase in pressure after the $P/P_0$ is greater than 0.18. This adsorption-pressure response is opposite to the general propensity for physical adsorption. The adsorption curves of both adsorbents are wrinkled more evidently than their desorption counterparts. Although PET-AC is somewhat more porous than MCR, the two specimens should possess similar porous structures sustained by the interstitial voids among the PAHs nanosheets because of their similar isotherm profiles.

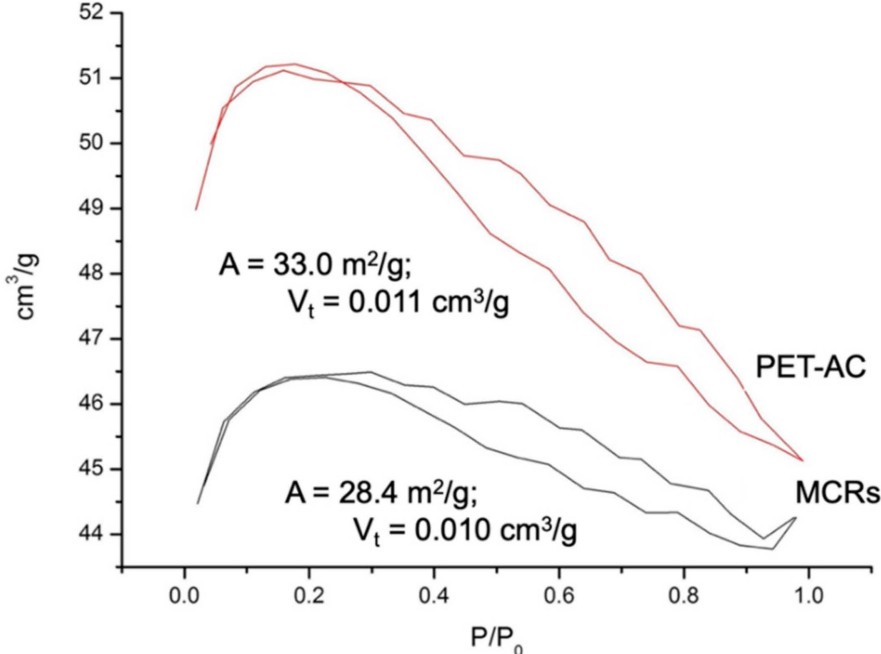

**Figure 6.** The comparison of the BET surface areas and total pore volumes of the two carbon specimens attained from pyrolysis of PET and annealing of the PET-AC in the co-gas, respectively.

Regarding the X-ray diffraction patterns, PET-AC and MCR exhibit the typical amorphous halo (Figure 7). The outcome indicates the lack of ordered assemblies of the curving PAH sheets to form MCR during the C-C transformation. This observation corresponds with the similar isotherms of both carbon specimens.

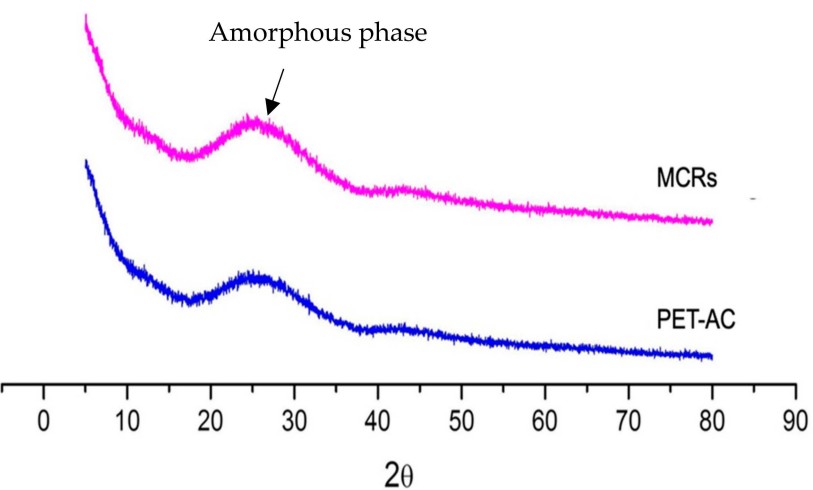

**Figure 7.** The X-ray diffraction patterns of the two carbon specimens in question.

## 4. Discussion

### 4.1. The Initial Stage That Leads to the Growth of Micro Carbon Rods (MCR)

Regarding how the formation of MCR is initiated, further inspection unveiled an appealing fact that when TEM zoomed in on a random residual AC bit in the inset of Figure 4. Hollow nanotubes made up entirely of multi-walled carbons (MWCNT) were observed (Figure 8a); they had grown forth from the AC nanograins (Figure 2). The MWCNT is generally formed through either CVD or catalytic choking process [26–29], but here was rather formed by a C-C transformation strategy. Therefore, it is deemed that the co-gas annealing brings about the curvature in PAHs, which initially pile up AC nanograins. This flat-to-bending happens presumably through the participation of $CO_2$ in the p-system through complicated reaction steps, such as addition and oxidation, under the co-gas atmosphere. The oxygen-containing species generated include ketenes and ethers (Figure 1). Adding the organic functional groups to the PAHs rings thus triggers the formation of tetrahedron sites or pentagon rings, resulting in curved PAHs [30]. In addition, helium in the co-gas treatment (Section 2.2) plays the role of a diluent to mitigate the oxidation power of $CO_2$. In principle, He is the lightest noble gas, least affecting, through molecular interactions, the reactivity of $CO_2$ with APH because of its weakest affinity with $CO_2$. Nevertheless, there are other choices that adjust the oxidizing power of $CO_2$ at high temperatures, e.g., using Ar or $N_2$ in relatively lower amounts vs. 50% of He in the co-gas.

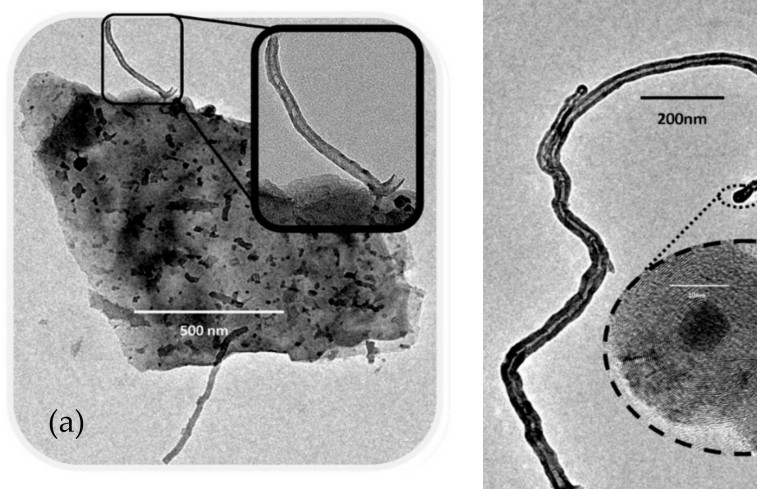

**Figure 8.** MWCNT growing forth directly from the carbonized PET-AC crumb during the C-C conversion (**a**); The detailed structure of the MWCNT (**b**).

Despite the fingerprint of IR (Figure 1) showing the organic functional groups of PET-AC and MCR, the formation of PAHs is also backed by the known fact that the pyrolysis of PET produces the graphitic carbon skeleton, according to a Raman spectroscopy study [31]. As previously mentioned, PET undergoes the thermal decomposition cascade at 450 °C; we carried out pyrolysis of PET eventually at 700 °C to attain PET-AC, over which amorphous carbon was formed and characterized by the aromatic C=C stretching absorption band (1597 cm$^{-1}$), as shown in Figure 1. On the contrary, MCR almost does not display a perceptible band of the aromatic C=C stretching, attributed to a lower ratio of the peripherical carbons over all sp$^2$ carbons. Correspondingly, the Raman spectrum must display D and G bands [32]. The conjugated sp$^2$ and sp$^3$ carbons are located primarily at PAH rings' inside parts and edges, respectively. Therefore, in principle, MCR must have a lower D/G peak intensity ratio than PET-AC.

The above assessment was furthered by observing a carbon nano node at the termination of the nanotubes, proposing that the carbon node was at first a nanograin, which had subsequently been consumed to supply the curved PAHs for the growth of the CNT, as illustrated in the following schematic (Figure 9). Although few discrete CNTs found are far thinner than MCR, the primary product of the C-C transformation, it is rational that the MCR started with those PAHs with relatively lower curvatures to form the initial cylindric prototypes for the afterward layer-by-layer stacking of the curved PAHs among nearby nanograins, leading to various MCR. It is important to stress that there is no spatial translation of PAHs in the co-gas stream, but rather a local translation (on a scale of a few microns) for the growth of MCR. The above assessment is based on Figures 4 and 8 using the mass balance concept. As described above, MCR has aspect ratios much larger than nanograins; hence, each MCR consumes a certain number of nanograins during annealing at 800 °C. It is the base temperature for carbonization to prepare carbon fiber [33]. Therefore, it is rational that PAHs underwent aromatic condensation to attain more extended sheets along the axial direction at this temperature. It is due to such random curvature stacking that the PAHs nanosheets in MCR lack long-range alignment and unveil the amorphous halo, seen in Figure 7. On the other hand, a substantially small number of PAHs with high curvatures end up with MWCNT.

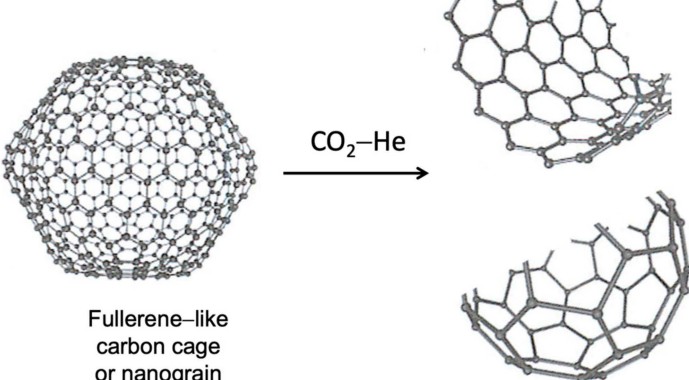

**Figure 9.** Schematic illustration of the curved PAH sheets' stripping from nano carbon cages or nanograins of AC.

Regarding the porous features of MCR as determined by the BET adsorption model (Figure 6 and Table 1), the inverted isotherms, as pointed out above, presumably originate from the adaptable micropores, namely the seams sandwiched by curved PAHs. The peeled carbon layer (inset of Figure 3) does not show porous morphology due to too-low amplification for visualizing the absorbent structure. As a consequence of inter-sheet spaces, the pore volume reduces with the increase in pressure (P/P$_0$) due to the squeezing effect. This interpretation is supported by a wrinkled shape of the adsorption curve; such a wrinkly profile reflects the PAH nanosheets' stepwise relaxation when the pressure is

applied. The adsorption curve of PET-AC is not so wrinkled because the wrinkled shape needs the prevailed sheet stacking layer structure and certain flexibility. Sample PET-AC has a somewhat greater surface area than sample MCR, rationally, because the former includes the contribution of the interstitial voids. Regarding the potential applications of MCR, besides functioning as the energy-damping filler in a plastic matrix, such as various rubber shock absorbers, they can also be used for reinforcing the mechanical strength of polymer composites because of their apt aspect ratios (>10) [34].

**Table 1.** The porous features of the two carbonaceous specimens.

| Sample | Surface Area ($m^2$/g) | $V_t$ ($cm^3$/g) | $V_m$ ($cm^3$/g) | $V_m/V_t$ |
|---|---|---|---|---|
| PET-AC | 33.0 | 0.011 | 0.0100 | 0.91 |
| MCR | 28.4 | 0.010 | 0.0095 | 0.95 |

Where $V_t$ is the total pore volume, $V_m$ is the volume of the micro-pores (<2 nm).

*4.2. How the P-Conjugated System Impacts Curvature Piling of PAHs*

Following the discussion on the co-gas treatment, one may consider the fine cylindrical structure of MCR. It is aware that it is amorphous according to the XRD examination, consistent with the two factors sustaining the rod growth; a nano carbon fiber, regardless of hollow or solid, is essential as the template, along with random stacking of the curved PAH sheets in nano sizes. Therefore, although the inset of Figure 4 displays a whole piece of the carbon layer on the micron scale, the layer must consist of numerous curled PAH nanosheets. On the contrary, the PPY-derived AC does not preserve sufficient reactivity to $CO_2$, the oxidizing agent, at the annealing temperature in the co-gas. Namely, no nano sheet curving happened in the PPy-AC, and hence no peeling off from the AC flakes occurred.

Regarding this, we used pure $CO_2$ to treat the PPy-AC; it turned out that none of the cylindrical carbon species was generated, but etching left behind numerous tiny ditches or cavities along the inter-particulate region (Figure 10) in contrast to the morphology of its intact counterpart shown in Figure 5. This observation implies that $CO_2$-driven oxidation occurs primarily at the edges of the highly stacked PAHs. These edges concentrate at the surface of the AC nanograins so that the $CO_2$-driven oxidation happens at the interparticle boundary regions. As for the vital structural difference between PET-AC and PPy-AC, the latter has a nitrogen-containing $sp^2$ carbon skeleton [35], and thus has the stronger association amid the PAHs, due to the permanent moments; as a result, it is difficult for the $CO_2$-driven oxidation to occur at the inner-ring carbon atoms.

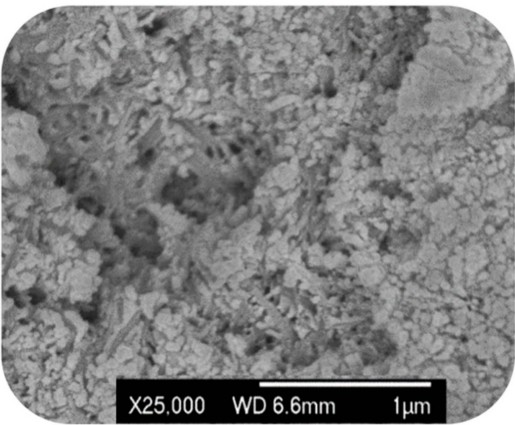

**Figure 10.** The morphology of PPy-AC after $CO_2$ etching.

*4.3. Future Improvement in the Technology to Prepare MCR from PET*

The C-C transformation of PET-AC to MCR is potentially attractive for producing carbon rods as structural or functional fillers from plastic PET waste through a more straightforward protocol. As elaborated above, forming nano cylindrical templates from

small-arched PAH species is the first step leading to MCR. Currently, Cn fullerene balls represent the arched PAH species, as illustrated by Figure 9. Regarding the path to forming fullerene balls in the pyrolysis stage, the approach relies on the condensation of small and curled PAHs and the boundary reaction to form C nuclei and subsequent growth. To enhance this path, the pyrolysis (Section 2.1) can be split into the decomposition at 400 °C and the $CO_2$ activation at 700 °C. A substantial grinding of the carbonaceous substances and a smaller amount of KOH (chemical etching) from the first step can prompt the reactive surface of carbon in $CO_2$ to generate far more Cn fullerene balls ready for the C-C transformation. An alternative to revamping the present protocol can be incorporating a small dosage of fullerene into PET-AC powder to produce the nano cylindrical templates. In addition, undertaking the C-C transformation in a fluidized bed reactor, instead of in the present fixed bed reactor, might increase the interlayer voids and pliability of the layered structure of MCR for applications as energy damping filler. The fluidization is deemed to render the curled PAHs able undertake more arbitrary stacking on the cylindrical templates. In the meantime, the fixed-bed reactor can be modified by changing the axial flow pattern to the radial flow pattern to enhance the aspect ratios and achieve a narrow distribution of them. This change is attained by mitigating the trans-tube pressure drop.

**5. Conclusions**

The present investigation reports an unusual way to attain micro carbon rods (MCR) from polyethylene terephthalate (PET), comprising pyrolysis in an inert atmosphere to obtain activated carbon (AC) and C-C transformation through annealing the AC in the He-$CO_2$ co-gas atmosphere. The C-C transformation experiences five mechanistic steps: (i) the oxidation of cyclic polyaromatic hydrocarbons (PAHs), formed from pyrolysis of PET by $CO_2$, to cultivate the curvature of PAH sheets; (ii) the assembling of the curved PAHs, leading primarily to MCR, which requires in the first place the formation of nano cylindrical carbon species, including the multiple-walled carbon nanotubes; (iii) the co-gas treatment for the C-C transformation employs He as a diluent to mitigate the oxidizing power of $CO_2$; (iv) the preparation of PET-AC to make the PAHs reactive enough for the generation of MCR has to be conducted first at a relatively mild temperature and then at a carbonization temperature in an oxidizing atmosphere ($CO_2$); and (v) the generation of MCR typically happens within some adjacent nanograins, in a very local manner, although more specific information on rod growth still needs investigating. In addition to the above remarks, it is appealing to observe that MCR species bear voids between various carbon layers, displaying an inverted BET adsorption isotherm profile with a wrinkly contour. This phenomenon reflects that interlayer spaces are pliable with the alternation of pressure; such structural features will likely make MRS a unique filler for absorbing vibration energy. Lastly, the MRS possesses applicable aspect ratios for mechanically reinforcing polymer composites.

**Author Contributions:** Conceptualization, Y.Z. and L.H.; methodology, Y.Z.; formal analysis and investigation, L.H.; resources, L.H.; data curation, Y.Z.; writing—original draft preparation, Y.Z.; writing—review and editing, Y.Z. and L.H.; visualization, L.H.; supervision, L.H.; project administration, L.H.; funding acquisition, L.H. All authors have read and agreed to the published version of the manuscript.

**Funding:** This research was funded by the NRF proof-of-concept grant scheme (Project title: Ceramic Pore-Channels with Inducted Carbon-nanotube Fence for Removing Oil from Water, Grant No.: NRF-POC-001-047).

**Acknowledgments:** The authors thank the National Research Foundation (NRF) of Singapore for funding this research.

**Conflicts of Interest:** The authors declare no conflict of interest.

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
