# Peer review of "Unusual Micro Carbon Rods Formed from PET Plastic via Pyrolysis and Annealing in CO2/He Co-Gas"

_jcs, doi:10.3390/jcs7050205_

Round 1

Reviewer 1 Report

In this work, Hong and Zhou prepared the carbon rods (MCR) derived from polyethylene terephthalate (PET) through pyrolysis, which was further characterized by FT-IR, PXRD, FE-SEM, BET. Electron microscopic characterization of MCR suggests a different growth mechanism from the sublimation and condensation, with the low-porous, amorphous, short micro characteristics. The results seem interesting and good, and can meet the requirement of J. Compos. Sci., thus in my opinion, this work can be accepted after careful revision according to the following detailed comments:

1.      The introduction needs to be improved. Is it based on the consideration of structure or related research support? Therefore, I suggest the author make some necessary modifications in the introduction.

2.      The author only discussed the preparation of carbon rods (MCR), so in my opinion, there should be at least relevant discussion on whether MCR have important potential application.

3.      Some thoughts and suggestions during the synthesis process are added to the Conclusions section to provide inspiration for readers

4.      To arouse a broad interest from the readership, several previous works in this field can be enriched, such as eScience, 2023, 2, 243-277; J. Rare Earths, 2022, 40, 1751.

5.      The clarity of all the pictures in the manuscript needs to be improved. The author can also check whether the pictures are clear enough.

6.      There are some grammar mistakes in this manuscript, the English of the manuscript should be further improved.

 There are some grammar mistakes in this manuscript, the English of the manuscript should be further improved.

Author Response

  1. The introduction needs to be improved. Is it based on the consideration of structure or related research support? Therefore, I suggest the author make some necessary modifications in the introduction.

Response: Thanks for the positive comment on this manuscript. The origin and support of this research project have now been described in the last paragraph of the Introduction.   

  1. The author only discussed the preparation of carbon rods (MCR), so in my opinion, there should be at least relevant discussion on whether MCR have important potential application.

Response: Thanks for this good point, which has been addressed in the last few lines of the Introduction and in the 3rd paragraph of Section 4.1. Typically, MCR could be used as a filler of composite for energy attenuation and structure reinforcement. 

  1. Some thoughts and suggestions during the synthesis process are added to the Conclusions section to provide inspiration for readers.

Response: Thanks for this comment; the corresponding amendment has been made to the Conclusion.

  1. To arouse a broad interest from the readership, several previous works in this field can be enriched, such as eScience, 2023, 2, 243-277; J. Rare Earths, 2022, 40, 1751.

Response: Thanks for this recommendation; we cite the second work above (ref. 17) as it is more related to our manuscript.

  1. The clarity of all the pictures in the manuscript needs to be improved. The author can also check whether the pictures are clear enough.

Response: Thanks for this comment; we have ensured that all the figures are clear.

  1. There are some grammar mistakes in this manuscript, the English of the manuscript should be further improved.

Response: Thanks for this helpful comment; we have used Grammarly to check the manuscript through this revision.

Reviewer 2 Report

The authors investigated a so-called unusual way to produce micro carbon rods from polyethylene terephthalate (PET), comprising pyrolysis in an inert atmosphere and C-C transformation in the He-CO2. The tools they used for characterization are FTIR, SEM, TEM, BET, and XRD.

From FTIR they observed functional groups grown on the carbon surface. They claimed they have observed carbon-carbon transformation. In a way, yes, but it is only limited to the surface, besides FTIR tells you the vibration information of the functional groups, not carbon structure transformation. They claim the huge background baseline change is due to carbon-carbon transformation. It is too early to tell.

They also claimed that from SEM and TEM they have observed the carbon-carbon transformation by saying that the production of tube-shaped objects (MWCNTs) is the evidence. Again, I’ll say, it’s too early to tell. From XRD, there is almost no difference, the broad peak is amorphous carbons, not MWCNTs.

The authors claimed that they have achieved carbon-carbon transformation based on the existence of several pieces of tube-shaped objects and the growth of functional groups on the carbon surface. Indeed, such a conclusion is unusual and uncertain.

The solution to the problems mentioned above is Raman spectroscopy. It tells you the true carbon-carbon transformation information. the near-infrared laser beam can penetrate several layers deep from the carbon surface to probe the inner carbon structures. this work is interesting and unusual. It lacks powerful evidence provided by Raman spectroscopy. It is not suitable for publication unless the Raman data is provided.

Author Response

  1. From FTIR they observed functional groups grown on the carbon surface. They claimed they have observed carbon-carbon transformation. In a way, yes, but it is only limited to the surface, besides FTIR tells you the vibration information of the functional groups, not carbon structure transformation. They claim the huge background baseline change is due to carbon-carbon transformation. It is too early to tell.

Response: Thanks for this valuable comment. To elucidate this point, we include a description of this point in the second paragraph on page 10.  

  1. They also claimed that from SEM and TEM they have observed the carbon-carbon transformation by saying that the production of tube-shaped objects (MWCNTs) is the evidence. Again, I’ll say, it’s too early to tell. From XRD, there is almost no difference, the broad peak is amorphous carbons, not MWCNTs.

Response: Thanks for pointing out this unclear aspect. We now carefully distinguished the carbon rods from NWCNTs and assigned the former to contribute to the amorphous XRD pattern.  

  1. The authors claimed that they have achieved carbon-carbon transformation based on the existence of several pieces of tube-shaped objects and the growth of functional groups on the carbon surface. Indeed, such a conclusion is unusual and uncertain.

 Response: We fully agree with this comment and aim to solicit more insights into this carbon-to-carbon transformation.

  1. The solution to the problems mentioned above is Raman spectroscopy. It tells you the true carbon-carbon transformation information. the near-infrared laser beam can penetrate several layers deep from the carbon surface to probe the inner carbon structures. this work is interesting and unusual. It lacks powerful evidence provided by Raman spectroscopy. It is not suitable for publication unless the Raman data is provided.

Response: Thanks for recommending us to carry out Raman spectroscopy characterization. Many studies have used Raman spectroscopy to date regarding the structural evolution of PAHs to graphitic carbon sheets. We elucidate this topic on page 10 by also citing two articles.

Reviewer 3 Report

Overall, the paper is interesting, but the following remarks need to be taken into account:

-   the introduction should be expanded also referring to the use of the obtained micro carbon rods in composites to fully fit with the “aims & scope” of the journal

-       pag 3: check the last sentence because it is incomplete

-   fig.3: the authors should provide some additional details or discussion because it is not possible to clearly distinguish cage balls/fullerenes from that micrograph magnification

-       pag. 8: check the references to the figures in the text (probably “Figure 5” should be replaced with Figure 6 and “Figure 3” with Figure 4)

-    the literature review presented in the manuscript is not exhaustive (it includes only 4 papers from the last 5 years) and needs to be expanded

Author Response

  1. The introduction should be expanded also referring to the use of the obtained micro carbon rods in composites to fully fit with the “aims & scope” of the journal

Response: Thanks for this valuable comment. The Introduction has been carefully revised to incorporate this aspect.

  1. Pag 3: check the last sentence because it is incomplete

Response: Thanks for pointing out this flaw. It has been corrected.

  1. 3: the authors should provide some additional details or discussion because it is not possible to clearly distinguish cage balls/fullerenes from that micrograph magnification.

Response: Thanks for this specific comment. We have amended the label added to Figure 3 and clarified the first paragraph on page 6.

  1. 8: check the references to the figures in the text (probably “Figure 5” should be replaced with Figure 6 and “Figure 3” with Figure 4)

Response: Thanks for pointing out this error. We have corrected it.

  1. the literature review presented in the manuscript is not exhaustive (it includes only 4 papers from the last 5 years) and needs to be expanded

Response: Thanks for pointing out this weak aspect. We have added 5 more papers published in the last five years.

Round 2

Reviewer 2 Report

The authors cunningly bypassed the request for adding Raman spectroscopy in their experimental methods of the paper. They claimed their result by getting evidence from the literature search. If the aim of this paper is engineering, it seems OK. 

Reviewer 3 Report

The manuscript was revised and it is now suitable for publication